# Anal heterosex among young people and implications for health promotion: a qualitative study in the UK

C Marston, R Lewis

Faculty of Public Health and Policy, London School of Hygiene & Tropical Medicine, London, UK

**Correspondence to**
Dr Cicely Marston;
Cicely.Marston@lshtm.ac.uk

## ABSTRACT

**Objective:** To explore expectations, experiences and circumstances of anal sex among young people.

**Design:** Qualitative, longitudinal study using individual and group interviews.

**Participants:** 130 men and women aged 16–18 from diverse social backgrounds.

**Setting:** 3 contrasting sites in England (London, a northern industrial city, rural southwest).

**Results:** Anal heterosex often appeared to be painful, risky and coercive, particularly for women. Interviewees frequently cited pornography as the 'explanation' for anal sex, yet their accounts revealed a complex context with availability of pornography being only one element. Other key elements included competition between men; the claim that 'people must like it if they do it' (made alongside the seemingly contradictory expectation that it will be painful for women); and, crucially, normalisation of coercion and 'accidental' penetration. It seemed that men were expected to persuade or coerce reluctant partners.

**Conclusions:** Young people's narratives normalised coercive, painful and unsafe anal heterosex. This study suggests an urgent need for harm reduction efforts targeting anal sex to help encourage discussion about mutuality and consent, reduce risky and painful techniques and challenge views that normalise coercion.

## INTRODUCTION

Anal sex is increasingly prevalent among young people, yet anal intercourse between men and women—although commonly depicted in sexually explicit media—is usually absent from mainstream sexuality education and seems unmentionable in many social contexts.

Surveys suggest that young men and women—and older adults—are engaging in anal intercourse more than ever before.[1–4] Sexually explicit media depictions are often mentioned as affecting how sex is viewed and practised by young people,[5–7] with anal intercourse being one of the 'high risk' practices thought to be promoted by such media,[8 9]

## Strengths and limitations of this study

- This study uses a large qualitative sample from three diverse sites in England and is the first to capture a wide range of circumstances around and reasons for engaging in anal sex among men and women between the ages of 16 and 18.
- Analysis explores experiences in depth, going beyond simplistic explanations linking motivations for anal sex with pornography.
- The study shows that young people's narratives about anal sex contained ideas normalising coercive, painful and unsafe anal sex. These ideas could be addressed in health promotion work.
- This study was conducted in England and further work is needed to assess the extent to which similar discourses operate among young people in other countries.

although evidence about the influence of pornography on anal practices is thin.[5] Studies of anal practices, which are generally of over-18-year-olds,[10–12] suggest that anal sex might be desired by young men more than women and may be used to avoid pregnancy,[12 13] or vaginal intercourse during menstruation,[12] while often being unprotected with condoms.[12–14] It may be painful for women,[12 13 15] and may be a pleasurable part of sex for men and women.[16 17] Almost one in five 16–24 year-olds (19% of men and 17% of women) reported having had anal intercourse in the past year in a recent national survey in Britain.[4]

Very little is known about the detailed circumstances around or reasons for engaging in anal sex among under-18-year-olds anywhere, or what implications these might have for health. This study looks in detail at anal practices among young people aged 18 and under, develops hypotheses for further study and makes suggestions for sexual health promotion.

## METHOD
### Design and data collection
The narratives about anal heterosex presented here emerged as part of a longitudinal, qualitative mixed methods study (the 'sixteen18' project) which explored the range and meaning of different sexual activities among a diverse sample of 130 young people aged 16–18 in three contrasting locations in England: London; a medium-sized northern industrial city and a rural area in the southwest. From January 2010, we conducted 9 group interviews and 71 depth interviews (wave one: 37 women and 34 men), re-interviewing 43 of the depth interviewees 1 year later (wave two), until June 2011. The London School of Hygiene and Tropical Medicine Research Ethics Committee approved the study and all participants provided written consent.

For the depth interviews, we used purposive sampling to maximise variation in social background. Within each location, we sampled from a range of settings including: schools/colleges; youth work services targeting young people not in education or training; youth organisations; a supported housing project for young people living independently from their families; and informal networks. We also used 'snowball' sampling and, in the rural southwest, we approached people directly in a town centre. The sample was diverse in terms of economic and social background, and less diverse in terms of ethnicity (most participants were white British). See Lewis et al[18] for further details. We highlighted in our information leaflet and our conversations with potential interviewees that we were keen to speak to any young person, whatever their experiences. Although participants varied in terms of the range of activities they had experienced, and the number and nature of their sexual partnerships, the majority reported opposite-sex partners only.

In the depth interviews, we asked interviewees about what sexual practices they had engaged in, the circumstances of those practices and how they felt about them. We deliberately left 'sexual practices' undefined, to allow for young people's own definitions to emerge. In the group discussions, we asked general questions about what practices they had heard of, their attitudes to those practices and whether they thought young people their age would generally engage in particular practices, and if so, under what circumstances. Many of our interviewees talked about anal sexual practices unprompted (whether they had engaged in them or not) and so in wave two, we specifically asked all of our participants about their perception and, if relevant, their experience of anal practices (about a quarter of our in-depth interviewees reported anal sexual experiences). Our aim was to explore the key discourses surrounding anal sexual practices among this age group and to elicit detailed accounts of specific experiences.

### Data analysis
We recorded and transcribed all interviews. We used iterative thematic analysis[19] to develop our understanding of the data. This involved 'coding' transcripts[19] and extensive discussions between researchers to come to a shared interpretation of young people's accounts of anal sex, taking into consideration our own characteristics (eg, white, middle-class women older than the interviewees) and how these may have affected the data collected. We made constant comparisons across cases and themes, and sought 'deviant cases' to challenge our emerging interpretations. Throughout the analysis, we simultaneously engaged with theoretical literature to put the work in context.

We use unique identifier pseudonyms throughout. Quotations are from one-to-one interviews unless otherwise indicated, with omissions marked […].

## RESULTS
Anal practices reported usually involved penetration or attempted penetration by the man with his penis or finger and, with one exception, were between opposite-sex partners. Anal practices generally occurred between young men and women in 'boyfriend/girlfriend' relationships. Although a small minority of interviewees said anal sex (ie, penetration with a penis) was exclusively 'gay', it was widely understood as also occurring between men and women.

Initial anal sexual experiences were rarely narrated in terms of mutual exploration of sexual pleasure. Women reported painful anal sex:

> As soon as the whole incident happened where he didn't warn me it just hurt. It was just pain [*laugh*]. It was just like: no. No one could possibly enjoy that. It was just horrible […] I guess he could have used lube, maybe that would have helped, but I don't know. Apparently if you're tense it hurts more, I guess, which makes sense really, but I don't see how you couldn't be tense [*laugh*] in that kind of situation. (Emma)

Young men in our study, while often keen on anal sex in principle, were sometimes unenthusiastic about the physical reality: "I thought it was going to be a lot better to be honest" (Ali); "sometimes it does feel better [than vaginal sex] but I wouldn't say I preferred it" (Max).

From the young people's accounts, it seems that condoms were not often used, and when they were it was usually for basic hygiene, not sexually transmitted infection (STI) prevention: "so you don't get shit on your dick" (Carl). Some interviewees incorrectly stated that anal STI transmission was impossible, or less likely than for vaginal intercourse.

There were marked gender differences in how anal sex was described: its benefits (pleasure, indicator of sexual achievement) were expected for men but not women; its risks—interviewees rarely mentioned risks of STIs, focusing instead on risk of pain or damaged reputation—were expected for women but not men. Our interviewees did not describe anal sex as a way to preserve virginity or avoid pregnancy.

## Reasons for anal sex

The main reasons given for young people having anal sex were that men wanted to copy what they saw in pornography, and that 'it's tighter'. The implication was that 'tighter' was better for men and was something men were said to want, while women were expected to find anal sex painful, particularly the first time. The 'pornography' explanation seems partial at best, not least because young people only seemed to see this as motivating men, not women. We found other important explanations and motivations in young people's accounts, as we will see below.

Key themes emerged from our interviews that help explain why the practice continued despite narratives of women's reluctance, expectations of pain for women and apparent lack of pleasure for women and men: competition between men; the claim that 'people must like it if they do it' (alongside the seemingly contradictory expectation that it will be painful for women); and—crucially—normalisation of coercion and 'accidental' penetration.

## Competition between men

While not all young men in the study wanted to have anal sex (eg, saying it was 'not for them'), many men said they encouraged one another to try the practice, and men and women said men wanted to tell their friends that they had had anal sex. Men in a group discussion said anal sex was 'something we do for a competition', and 'every hole's a goal'. By contrast, men and women said women risked their reputation for the same act, a sexual double standard familiar from previous literature.[20]

## People must like it if they do it

Despite asserting that anal sex is inevitably painful for women, and despite not usually linking pain to any sexual pleasure, men and women often also expressed the seemingly contradictory view that anal sex was in fact enjoyable for women:

> Obviously people do enjoy it if they do it. (Naomi)

> There's quite a few, a lot of girls enjoy it. But I think most girls would like, I think they might do, on the quiet. (Shane)

That it 'must' be enjoyable was typically suggested as an explanation by those who had not engaged in the practice.

Women experiencing pain were often depicted as naive or flawed. Men and women said that women needed to 'relax' more, to 'get used to it':

> I think that the boy enjoys it. I think it's definitely the boy that pushes for it from watching porn and stuff, they wanna try it. The girl is scared and thinks it's weird, and then they try it because the boyfriend wants them to. *They normally don't enjoy it because they're scared* and I, I know that like with anal, *if you're not willing, you don't relax,*

> like if you have, you have control over two of the muscles that are closest to the outside and then inside it's like involuntary and if you're scared or you haven't eased them off like they stay tight and then you can rip 'em if you try and force anal sex. (Mark [our emphasis])

Note that Mark refers, almost casually, to the idea that a woman might be 'scared' or 'not willing' in a scenario in which anal sex is possibly taking place, seemingly assuming a shared understanding with the interviewer that this would often be the case. Elsewhere in the interview, he talks about having hurt his partner during an anal sex 'slip' (see below), and so his talk about 'easing off' may reflect his own—perhaps more recent—understanding of how it 'should' be performed.

## Normalisation of coercion and 'accidental' penetration

The idea that women would generally not wish to engage in anal sex, and so would need to be either persuaded or coerced, seemed to be taken for granted by many participants. Even in otherwise seemingly communicative and caring partnerships, some men seemed to push to have anal sex with their reluctant partner despite believing it likely to hurt her (although it should also be noted that other men said they avoided anal sex because they believed it might hurt their partners). Persuasion of women was a feature to a greater or lesser degree of most men's and women's narratives about anal sex events, with repeated, emphatic requests from men commonly mentioned.

Women seemed to take for granted that they would either acquiesce to or resist their partners' repeated requests, rather than being equal partners in sexual decision-making. Being able to say 'no' was often cited by the women as a positive example of their control of the situation.

Some men also described taking a 'try it and see' approach, where they anally penetrated a woman with their fingers or penis and hoped that she would not stop them.

Shane told us if a woman said 'no' when he started "putting [his] finger in", he might keep trying: "I can be very persuasive […]. Like sometimes you just keep going, just keep going till they just get fed up and let you do it anyway".

'Try it and see' generally either hurt the woman or was 'unsuccessful' (from the man's point of view) in the sense of not penetrating 'it just didn't go in really'. (Jack) A verbal 'no' from the woman did not necessarily stop anal penetration attempts:

> He tried putting it there.

> [Interviewer] Right

> And I just said 'no'.

> [Interviewer] Had he asked you first or did he just try it?

Um, he kept asking me at first. I'm like 'no', but then he tried it and I said 'no way'.

[Interviewer] Right

'No chance'. (Molly)

In some cases, anal penetration of the woman—digital or penile—was described by men and women as having happened accidentally ('it slipped'). For instance, Mark, mentioned above, told us about a time when he 'slipped' during a vagina-penis intercourse and penetrated his girlfriend anally.

Owing to the nature of the data—we rely on reports at interview—it is difficult to assess the extent to which events described as 'slips' were genuinely unintentional. One man, however, described a 'slip' at the first interview, which he said to the interviewer—and said he had told his girlfriend—was an accident, an account which he amended at the second interview:

[Interviewer] I think you said […] in the first interview that there had been a time where […] you said it [his penis] slipped.

Well I, I tried, and I said it slipped.

[Interviewer] So it hadn't actually slipped? It wasn't an accident?

No, no, no it wasn't an accident. (Jack)

Describing events as 'slips', then, may enable men and women to gloss over the possibility that penetration was deliberate and non-consensual.

The narratives suggested little expectation that young women themselves would want anal sex. Many young men, on the other hand, clearly described wanting to penetrate a woman anally. This mismatch may help explain why 'slips' and 'persuasion' of the woman were common features of the narratives about anal sex.

## Anal sex and pleasure

Among those who had had anal sexual experiences, few of the men and only one woman among this young age group referred to physical pleasure in their accounts. Alicia, the only woman narrating pleasurable anal penetration, exemplifies some of the complexities involved in women's navigating (and narrating) anal sexual practices. She described a fairly common pattern: her partner asked for anal sex, which she first refused but later agreed to. She found it painful, and also had a second experience where her consent to anal penetration was questionable ('it just kind of slipped in'). She was atypical, however, in that she related the story in a positive way emphasising her own agency ('I was curious about it') and described how she had subsequently enjoyed anal sex, suggesting that they had found a mutually satisfactory way to engage in the practice.

Her partner had had anal sex before. The first time she had anal sex with him was 'really painful':

I didn't wanna try it [anal sex] initially, well I was unsure about it initially. But I kind of, he didn't, he said 'that's fine', but I still wanted to try it for him because I was interested. I think I was interested to why he was interested. I was curious about it […] So I think that's […] I just sort of tried it for him.

She described the second occasion they had anal sex differently in the first and second interviews:

[First interview] We were having [vaginal] sex another time and it [his penis] just kind of slipped in [into her anus] that way.

[Second interview] He just sort of slipped in […] I think he thought it would make it less painful for me. And I think he thought he can make me like it like that.

At the first interview, Alicia was ambiguous about what happened, narrating the event as though it were accidental ('it just kind of slipped in'), perhaps reluctant to draw attention to not having been involved in the decision. At the second interview, she was clearer that he had deliberately penetrated her (she may also have spoken to her partner about it between interviews). She presents it in a somewhat positive way ('he thought he can make me like it') but her consent remains unclear.

At both interviews, she emphasised how much she enjoyed subsequent anal sex with the same man, and that either of them might initiate it. Alicia was the only woman we interviewed who described experiencing pleasure, including orgasm, from anal sex.

Yeah. I quite like it because I think I quite like the feeling of him against my bum, like against the meat of your bum, like it's kind of cushiony. So yeah, I think that's what I like about it, I'm not sure.

Alicia's case was also unusual in how she presented herself in relation to her partner as more sexually driven: "I'm not saying that I'm like wanting sex [all practices, not only anal sex] all the time, but I'd say I go for it more. I'd initiate it more".

In a previous work, we have shown how interpretations of apparently coercive events can change over time[21] and it is possible that better, later experiences in the context of a continuing relationship had allowed her to incorporate the initial, less enjoyable ones into a narrative of personal sexual growth within a stable relationship, particularly as she came to enjoy the practices that she had found painful at first.

Despite being generally positive, Alicia's account also contains indications of reluctance ("I didn't wanna try it […] I was unsure"). It is possible that even as she talks about enjoying the practice, her narrative was shaped to some extent by social expectations about women resisting anal sex. Similarly, men did not spontaneously talk about

not enjoying anally penetrating a woman, only mentioning it after direct questions, supporting other works describing an onus on men to articulate only a positive view of sex.[22 23]

## DISCUSSION

Few young men or women reported finding anal sex pleasurable and both expected anal sex to be painful for women. This study offers explanations for why anal sex may occur despite this.

Interviewees frequently cite pornography as the 'explanation' for anal sex, yet only seem to see this as a motivation for men. A fuller picture of why women and men engage in anal sex emerges from their accounts. It seems that anal sex happens in a context characterised by at least five specific features linked with the key explanatory themes described above:

First, some men's narratives suggested that mutuality and consent for anal sex were not always a priority for them. Interviewees often spoke casually about penetration where women were likely to be hurt or coerced ("you can rip 'em if you try and force anal sex"; "you just keep going till they get fed up and let you do it anyway"), suggesting that not only do they expect coercion to be part of anal sex (in general, even if not for themselves personally), but that many of them accept or at least do not explicitly challenge it. Some events, particularly the 'accidental' penetration reported by some interviewees, were ambiguous in terms of whether or not they would be classed as rape (ie, non-consensual penetration), but we know from Jack's interview that 'accidents' may happen on purpose.

Second, women being badgered for anal sex appears to be considered normal.

Third, the commonly circulating ideas that 'everyone' enjoys it, and that women who do not are either flawed or simply keeping their enjoyment secret, help support the erroneous idea that a man pushing for anal sex is simply 'persuading' his partner to do something that 'most girls would like'. Even Alicia's narrative contains some of the apparently coercive features of anal sex that other women report in negative terms, despite Alicia reporting enjoying anal sex.

Fourth, anal sex today appears to be a marker of (hetero)sexual achievement or experience, particularly for men.[18] The society which our interviewees inhabit seems to reward men for sexual experience per se ('every hole's a goal') and, to some extent, rewards women for compliance with sexually 'adventurous' acts (enjoyment signifying not being naive, unrelaxed, etc), although women must balance this with the risk to their reputation. Women may also be under pressure to appear to enjoy or choose certain sexual practices: Gill describes a 'postfeminist sensibility' in contemporary media, where women are expected to present themselves as having chosen behaviours that conform to a stereotype of heterosexual male fantasy.[24] The common portrayal of anal heterosex in terms of men breaking women's resistance can be compared with narratives about first vaginal intercourse[25] and perhaps have superseded them to some degree in the British context where premarital vaginal intercourse is considered normal and so perhaps less of a 'conquest'.

Fifth, many men do not express concern about possible pain for women, viewing it as inevitable. Less painful techniques (such as slower penetration) were rarely discussed.

Currently, this apparently oppressive context, and indeed the practice of anal heterosex itself, appears to be largely ignored in policy and in sexuality education for this young age group. Attitudes such as the inevitability of pain for women, or social failure to recognise or reflect on potentially coercive behaviour, seem to be unchallenged. Alicia's case demonstrates how women might absorb potentially negative experiences into an overall narrative of control, desire and pleasure, all of which she emphasises in her account.

We do not suggest that mutually pleasurable anal practices are not possible among this age group, nor that all men want to coerce their partners. Rather, we wish to emphasise how mutuality and women's pleasure are often absent in narratives of anal heterosex and how their absence is not only left unremarked and unchallenged, but even seems to be expected by many young people.

Previous work has suggested that gendered power may operate differently for different sexual activities, and that sexual 'scripts' (eg, expectations about how practices will be initiated and performed) for anal intercourse may not be as well established as for vaginal intercourse.[13] Our findings suggest that coercion could emerge as a dominant script for anal intercourse at these young ages if left unchallenged.

Further work is needed to assess the extent to which similar coercive discourses operate among young people in other countries. This is a qualitative study, with an in-depth analysis of a smaller sample than would be usual for epidemiological studies, but which spans three locations and diverse social groups. Whether or not the concept of 'generalisability' should be applied in qualitative research is a matter of debate,[26] but we would argue that this study provides useful, credible working hypotheses or theories about anal sexual practice among young men and women that are likely to apply outside our group of interviewees.

Sexuality education, and specifically what it should contain, is the subject of global debate.[27 28] Prevention of STIs, HIV and violence are priorities for health promotion worldwide. Yet sexuality education, where it exists, rarely addresses specific sexual practices, such as anal sex between men and women—despite its potential for disease transmission and, as these accounts reveal, coercion. In England, where this study was located, discussions of pleasure, pain, consent and coercion are included in good sexuality education but such education remains isolated, ad hoc and non-compulsory.

## CONCLUSION

Anal sex among young people in this study appeared to be taking place in a context encouraging pain, risk and coercion. Harm reduction efforts targeting anal sex may help encourage discussion about mutuality and consent, reduce risky and painful techniques and challenge views that normalise coercion.

**Acknowledgements** The authors thank Kaye Wellings and Tim Rhodes for their role in the project design, the two reviewers for their contribution, and Amber Marks and Crofton Black for their comments on an earlier draft of the manuscript.

**Contributors** CM and RL contributed to the planning, conduct and reporting of the work described in the manuscript. CM is the guarantor for this manuscript.

**Funding** Funding for this study was obtained from the Economic and Social Research Council (UK) RES-062-23-1756.

**Competing interests** None.

**Provenance and peer review** Not commissioned; externally peer reviewed.

**Ethics approval** Ethical approval was granted by the London School of Hygiene & Tropical Medicine Research Ethics Committee (Application #5608). All participants gave informed consent prior to taking part in this research.

**Data sharing statement** No additional data are available.

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
