## [Reviewer comments · BMJ Open]

Some articles will have been accepted based in part or entirely on reviews undertaken for other BMJ Group journals. These will be reproduced where possible.

ARTICLE DETAILS

TITLE (PROVISIONAL)	Anal heterosex among young people and implications for health promotion: a qualitative study in the UK
AUTHORS	Marston, Cicely; Lewis, Ruth

VERSION 1 - REVIEW

REVIEWER	Lesley Hoggart The Open University, UK
REVIEW RETURNED	21-Mar-2014

GENERAL COMMENTS	This is a really interesting, well-written paper in an area of sexual health for which there is little research evidence. Surveys (UK and USA) have indicated an increase in the practice of anal sex among young people, but little is known of the contexts and experiences of these practices. This qualitative study is therefore very welcome. It not only provides insights into the circumstances of this sexual behaviour but suggests policy responses. There are, however, some areas which would benefit from some revisions, primarily in order to clarify the points being made. Abstract. This is necessarily word limited, but I think it requires a bit more content. l.29 - 'anal sex appeared to be ...' - for both parties? This is developed in the paper but the question is left hanging here. Best to use semi-colons in results section. Abstract conclusions feel rather ambitious. Who is going to talk about anal sex with young people? (suggestion later in the paper is that this could be covered in sexuality education). Maybe this could be suggested as something to aim towards but with some other suggestions that may be more achievable ie. improving discussions around consent and coercion, so that young people can talk about 'not letting someone do stuff that I don't want to do, or which hurts me'; and maybe leaflets on anal sex at sexual health clinics etc. l.53 Not sure that 'promote' is the right word. It seems quite an obvious point that views that promote coercion should be challenged; maybe 'condone' or 'normalise'. p.4, l. 23. The lit review is very dense. Anal sex may be pleasurable just for men, or does the literature suggest women also? p.4. l.31. Drawing out global lessons for sexual health promotion sounds rather ambitious given you have just stated that further work is needed to assess the extent to which the same discourses operate in other countries. Maybe you are developing hypotheses for further study, with some suggestions for sexual health promotion. p.4 l.52. Variation in sample is mentioned but there is no information on this variation apart from geography. Was a diverse sample
--

	achieved? Results. p.6. It is not clear why numbers are given in a qualitative study. No claim to representation is made, so the numbers are misleading. p. 6-7. The first few of the key themes read rather like a list with important themes attracting little attention. It appears that these are setting the scene for themes that are considered to be more important (people must like it and normalisation of coercion). In addition, is the normalisation theme a distinct theme or an overarching theme? Maybe it would be better to focus on people must like it and normalisation of coercion themes? (see also comments for page 10. p.7 l.38. male or female interviewees? p.7 l.38-39. Can pain also be enjoyable? p.8 l.9. I think you need to define what you mean by coercion - persuasion/coercion can be a grey area. p.9. l.6. Would be good to have an example of when it is difficult to assess extent to which 'slips' are unintended. p.9 l.32 complexities: the paper has tended to read so far as though anal sex was a simple matter of men pressuring/women resisting; so it could maybe be a little more nuanced earlier on. And what are the complexities - could this be drawn out more? Was there no sense in which the women were willing, even excited, to experiment but then maybe it didn't work out for them? etc. Discussion: p.10. It is a bit confusing because these are another set of 5 themes (not the same themes). Are these related to the two final key themes only? p.10 l.57 and p.11. l. 3 Not sure about repeating quotes here. They are definitely out of context and don't seem 'casual'. p.11 p.38-39 (and also the Mark quote on p.7-8). It needs making clear that this is your interpretation of the quote in the context of the rest of the interview. On its own, another interpretation could be that Mark is thinking about his partner's enjoyment and would wish to avoid hurting her - eg. is he proposing that easing off is a less painful technique? p.11. On sexuality education, as mentioned previously maybe it would be helpful to locate the starting point as a discussion on pleasure/pain and consent/coercion. p.12 l.10 What is the debate? A reference or two here would help.
--	--

REVIEWER	Roger Ingham Centre for Sexual Health Research University of Southampton Southampton
REVIEW RETURNED	25-Mar-2014

GENERAL COMMENTS	This article presents some important data on a relatively ignored area in sexual health research, and makes a contribution to the literature. I do, however, have some reservations as it stands at the moment, which I feel need to be addressed prior to possible publication. Some of the reservations arise from the general dilemma inherent in reporting qualitative research data within the confines of a short article, such that much of the nuanced and contextual information is necessarily omitted. This has a number of effects, including: 1 a tendency to focus on the act itself with relative ignoring of the relationship; for example, Alicia reports her changing views on the activity as her relationship progresses, as well as her presumably becoming more relaxed with time. The 'slipped in' discourse (in the
--

absence of any seeming anger or resentment) may reflect her own reluctance to admit to willingness of what may be regarded as a taboo activity in her social milieu. This is the only case presented which refers to an actual relationship, but even then it does not make any reference to the possibly symbolic function (sharing something unusual and special) of anal sex within specific relationships. The positivity displayed (despite the initial painful initiation) is at odds with the general conclusions of the rest of the article.

2 it is not always clear whether the views cited come from young people who have attempted or managed the activity, or merely heard about it, watched it, or been told about it. For example, the quotes on page 4 seem to be based on hearsay and it is not clear how helpful these actually are in understanding the reality.

3 although we are told that 19 of the 71 participants had engaged in (or at least attempted) one form of anal penetration or another, there is no sex breakdown reported. And although it is well accepted that qualitative research cannot be used for exploring prevalence, some idea of what is known about prevalence in this age group from other studies (NATSAL3, for example) would be helpful. Linked to this, if a man attempted penetration, was told by his partner that it hurt, and then stopped trying, what category would this fall into? If he carried on trying despite her protestations, then the term 'rape' can presumably be legitimately used – but how many actual cases of such coercion were reported, and by men or women or both?

4 was there any indication as to why the vast majority of participants had not attempted anal activity? In terms of designing an educational activity, such contrasting material can be very helpful.

5 I wondered throughout the article to what extent the same issues of coercion and 'it slipped' might arise in a study on first experiences of vaginal sex also. In other words, are the authors claiming special status for anal sex, or merely reporting on the age-old patterns of pressure by men towards women in a new manifestation (which is not, of course, to imply it is at all acceptable)?

6 although the numbers were necessarily low, were there any tentative patterns between the different locations or sources of participant? Given what is known about gender roles and social backgrounds, was there any evidence of greater coercion, for example, amongst certain sub-groups?

7 were there any examples of young people choosing not to try anal activity because they had indeed seen it in pornography? In other words, could watching porn be educative in a risk-reducing direction?

8 the small sample and somewhat brief descriptive data presented make it hard to be convinced that there is indeed a dominant script emerging. Certainly, coercion appears to be an important script, and clearly warrants close attention (but then it always has been an issue in relationships), but the overall conclusions seem rather too strong on the basis of the data that have been reported.

In sum, whilst I fully support the conclusions that greater attention in sex and relationships education needs to be directed to issues of consent, coercion, etc. I fear that this article as it stands may serve to over-simplify what is really a quite complex situation.

Please take these as constructive comments, which they are very much intended to be.

VERSION 1 – AUTHOR RESPONSE

Reviewer: 1

Reviewer Name Lesley Hoggart

Institution and Country The Open University, UK

Please state any competing interests or state 'None declared': none declared

This is a really interesting, well-written paper in an area of sexual health for which there is little research evidence. Surveys (UK and USA) have indicated an increase in the practice of anal sex among young people, but little is known of the contexts and experiences of these practices. This qualitative study is therefore very welcome. It not only provides insights into the circumstances of this sexual behaviour but suggests policy responses.

There are, however, some areas which would benefit from some revisions, primarily in order to clarify the points being made.

Abstract. This is necessarily word limited, but I think it requires a bit more content.

Abstract amended as below.

I.29 - 'anal sex appeared to be ...' - for both parties? This is developed in the paper but the question is left hanging here.

Clarification added

Best to use semi-colons in results section.

Done and abstract amended according to suggestions below about presentation of the themes.

Abstract conclusions feel rather ambitious. Who is going to talk about anal sex with young people? (suggestion later in the paper is that this could be covered in sexuality education). Maybe this could be suggested as something to aim towards but with some other suggestions that may be more achievable ie. improving discussions around consent and coercion, so that young people can talk about 'not letting someone do stuff that I don't want to do, or which hurts me'; and maybe leaflets on anal sex at sexual health clinics etc.

We have amended the abstract so that it is worded as per the conclusions.

I.53 Not sure that 'promote' is the right word. It seems quite an obvious point that views that promote coercion should be challenged; maybe 'condone' or 'normalise'.

Changed to 'normalise'

p.4, I. 23. The lit review is very dense. Anal sex may be pleasurable just for men, or does the literature suggest women also?

Women also - we have clarified

p.4. I.31. Drawing out global lessons for sexual health promotion sounds rather ambitious given you have just stated that further work is needed to assess the extent to which the same discourses

operate in other countries. Maybe you are developing hypotheses for further study, with some suggestions for sexual health promotion.

We have amended as suggested.

p.4 l.52. Variation in sample is mentioned but there is no information on this variation apart from geography. Was a diverse sample achieved? Results. p.6. It is not clear why numbers are given in a qualitative study. No claim to representation is made, so the numbers are misleading.

We have added some more information on the sample composition and have amended the text to avoid inadvertently implying that our qualitative data sampling equates to a statistical sample.

p. 6-7. The first few of the key themes read rather like a list with important themes attracting little attention. It appears that these are setting the scene for themes that are considered to be more important (people must like it and normalisation of coercion). In addition, is the normalisation theme a distinct theme or an overarching theme? Maybe it would be better to focus on people must like it and normalisation of coercion themes? (see also comments for page 10).

We have highlighted the main themes, and have merged the remainder into an introductory paragraph. We have referred back to the main themes in the discussion.

p.7 l.38. male or female interviewees?

We have clarified.

p.7 l.38-39. Can pain also be enjoyable?

We have added a note to clarify the context of this discussion.

p.8 l.9. I think you need to define what you mean by coercion - persuasion/coercion can be a grey area.

This is a key point, and we have previously written on the difficulties of defining coercion, particularly with respect to interview data. (Marston C. What is heterosexual coercion? *Interpreting narratives from young people in Mexico City. Sociology of Health & Illness. 2005;27(1):68–91.*) We have rephrased this section.

p.9. l.6. Would be good to have an example of when it is difficult to assess extent to which 'slips' are unintended.

We have clarified that we were referring to the use of interview data.

p.9 l.32 complexities: the paper has tended to read so far as though anal sex was a simple matter of men pressuring/women resisting; so it could maybe be a little more nuanced earlier on. And what are the complexities - could this be drawn out more? Was there no sense in which the women were willing, even excited, to experiment but then maybe it didn't work out for them? etc.

The only woman who was positive about anal sex was 'Alicia'. We have added emphasis in the text about the young age of these men and women because it seems likely that some of them would go on to try anal sex (possibly under more mutual arrangements) at older ages. We have also added more detail about the complexities here and elsewhere in the paper and have noted that narratives of anal sex are also gendered.

Discussion: p.10. It is a bit confusing because these are another set of 5 themes (not the same themes). Are these related to the two final key themes only?

We have linked the discussion more clearly with the preceding section.

p.10 l.57 and p.11. l. 3 Not sure about repeating quotes here. They are definitely out of context and don't seem 'casual'.

We have justified the 'casual' comment in the first appearance of this quote and have added some context. It was very normal for men to refer to anal sex hurting women and to talk about how women would resist or be scared or tense.

p.11 p.38-39 (and also the Mark quote on p.7-8). It needs making clear that this is your interpretation of the quote in the context of the rest of the interview. On its own, another interpretation could be that Mark is thinking about his partner's enjoyment and would wish to avoid hurting her - eg. is he proposing that easing off is a less painful technique?

We have now provided more contextual information about this to explain our interpretation.

p.11. On sexuality education, as mentioned previously maybe it would be helpful to locate the starting point as a discussion on pleasure/pain and consent/coercion.

We have added text to this effect.

p.12 l.10 What is the debate? A reference or two here would help.

We have added a citation to a paper that summarises the debate.

Reviewer: 2

Reviewer Name Roger Ingham

Institution and Country Centre for Sexual Health Research
University of Southampton
Southampton
SO17 1BJ
UK

Please state any competing interests or state 'None declared': none declared

This article presents some important data on a relatively ignored area in sexual health research, and makes a contribution to the literature. I do, however, have some reservations as it stands at the moment, which I feel need to be addressed prior to possible publication.

Some of the reservations arise from the general dilemma inherent in reporting qualitative research data within the confines of a short article, such that much of the nuanced and contextual information is necessarily omitted. This has a number of effects, including:

1 a tendency to focus on the act itself with relative ignoring of the relationship; for example, Alicia reports her changing views on the activity as her relationship progresses, as well as her presumably becoming more relaxed with time. The 'slipped in' discourse (in the absence of any seeming anger or

resentment) may reflect her own reluctance to admit to willingness of what may be regarded as a taboo activity in her social milieu. This is the only case presented which refers to an actual relationship, but even then it does not make any reference to the possibly symbolic function (sharing something unusual and special) of anal sex within specific relationships. The positivity displayed (despite the initial painful initiation) is at odds with the general conclusions of the rest of the article.

We have clarified that the anal sexual practices generally occurred within 'boyfriend/girlfriend' relationships and have added a citation to another paper where we discuss how different practices are linked with different relationships.

The point about sharing something special did not appear in our data.

We now emphasise how our 'atypical' case illustrates how women can take control of their sexual lives and how in the context of overall control over their lives, can absorb potentially negative experiences into an overall narrative of control, desire and pleasure, all of which she emphasises in her account. We have added a note about the possibility of pleasure through mutually agreed anal practices, and have also mentioned the additional layer of pressure on young women to agree to anal sex which may come from media and other depictions of women where women are principally valued if they are 'sexy and up for it' (citing Ros Gill's work). We have also added a note about gendered narratives of anal sex. Although in Alicia's case we do not think that there is evidence she was reluctant to talk about her enjoyment/willingness, we do agree that accounts of anal sex are gendered and framed accordingly which may help explain her remarks about being initially unwilling.

2 it is not always clear whether the views cited come from young people who have attempted or managed the activity, or merely heard about it, watched it, or been told about it. For example, the quotes on page 4 seem to be based on hearsay and it is not clear how helpful these actually are in understanding the reality.

We have now clarified in the text that the stereotypical views we report were generally expressed by participants who had not experienced anal practices.

3 although we are told that 19 of the 71 participants had engaged in (or at least attempted) one form of anal penetration or another, there is no sex breakdown reported. And although it is well accepted that qualitative research cannot be used for exploring prevalence, some idea of what is known about prevalence in this age group from other studies (NATSAL3, for example) would be helpful.

We have added figures from Natsal 3 into the introduction.

3 (continued) Linked to this, if a man attempted penetration, was told by his partner that it hurt, and then stopped trying, what category would this fall into? If he carried on trying despite her protestations, then the term 'rape' can presumably be legitimately used – but how many actual cases of such coercion were reported, and by men or women or both?

We have added a definition of rape (i.e. penetration without consent) into the text. We do not suggest pain per se indicates that the act is non-consensual. We do not attempt to define any specific act reported to us as 'rape'.

We explain in the text that we cannot know for sure from our interviews whether or not sex was consensual, but note that in some cases consent is in doubt (we give the example where a young man tells us he attempted to penetrate his unwilling partner and then told her it was a 'slip'.) Part of the purpose of this paper is to draw attention to language used to gloss over lack of consent e.g. 'slips'. The women who suffer 'slips' had not been able to give explicit consent because they did not

know what was about to happen.

4 was there any indication as to why the vast majority of participants had not attempted anal activity? In terms of designing an educational activity, such contrasting material can be very helpful.

Many had not attempted anal sex, but as we mention in the text, many young men said they would not rule it out in the future. Also some of our female interviewees said they would not do it at first interview, but then subsequently had done by second interview. Among those who had not had anal sexual experiences, it would be difficult to distinguish between those who chose not to have anal sex, those who had not been coerced into doing it, and those who had not had the opportunity to do it.

5 I wondered throughout the article to what extent the same issues of coercion and 'it slipped' might arise in a study on first experiences of vaginal sex also. In other words, are the authors claiming special status for anal sex, or merely reporting on the age-old patterns of pressure by men towards women in a new manifestation (which is not, of course, to imply it is at all acceptable)?

We have added a note into the discussion about coercion with respect to vaginal sex. It seems that 'slips' in our accounts are particular to anal sex, perhaps because of the mechanics: slipping from vaginal to anal sex may be more plausible than slipping from non-penetration to coitus?

6 although the numbers were necessarily low, were there any tentative patterns between the different locations or sources of participant? Given what is known about gender roles and social backgrounds, was there any evidence of greater coercion, for example, amongst certain sub-groups?

Because gender emerged as the pre-eminent category shaping narratives of anal heterosex for our sample, that is what we have focused on in the paper. Narratives of coercion were present in accounts across the sample, from members of diverse social and ethnic groups, and there was no evidence that expectations of coercive anal sex were more dominant in certain groups. It would be interesting to analyse different ways of talking about coercion in different sub-groups, but this is beyond the scope of the current paper.

7 were there any examples of young people choosing not to try anal activity because they had indeed seen it in pornography? In other words, could watching porn be educative in a risk-reducing direction?

There were no specific examples of this.

8 the small sample and somewhat brief descriptive data presented make it hard to be convinced that there is indeed a dominant script emerging. Certainly, coercion appears to be an important script, and clearly warrants close attention (but then it always has been an issue in relationships), but the overall conclusions seem rather too strong on the basis of the data that have been reported.

We have rephrased the conclusions.

In sum, whilst I fully support the conclusions that greater attention in sex and relationships education needs to be directed to issues of consent, coercion, etc. I fear that this article as it stands may serve to over-simplify what is really a quite complex situation.

Please take these as constructive comments, which they are very much intended to be.

The complexities have sometimes been edited out of the text in our attempt to make the paper suitable for a medical rather than a sociological journal. We have added more detail back in, in response to both reviewers' thoughtful comments.

VERSION 2 – REVIEW

REVIEWER	Lesley Hoggart The Open University UK
REVIEW RETURNED	23-May-2014

GENERAL COMMENTS	I am happy with the revised paper, and think it makes a very important contribution to the literature.
--

REVIEWER	Roger Ingham Centre for Sexual Health Research University of Southampton UK
REVIEW RETURNED	05-May-2014

GENERAL COMMENTS	A much clearer and balanced paper after taking account of the initial reviewer comments. You may wish to consider adding the reviewers to the names of people to thank for commenting on the paper - it is greatly improved as a result of the process! Well done.
--